# Aromatase deficiency in hematopoietic cells improves glucose tolerance in male mice through skeletal muscle-specific effects

**Katya B. Rubinow**[1]*, **Laura J. den Hartigh**[1], **Leela Goodspeed**[1], **Shari Wang**[1], **Orhan K. Oz**[2]

**1** Division of Metabolism, Endocrinology, and Nutrition, Department of Medicine, University of Washington, Seattle, Washington, United States of America, **2** Department of Radiology, University of Texas – Southwestern, Dallas, Texas, United States of America

* rubinow@uw.edu

**Data Availability Statement:** All relevant data are within the manuscript file.

**Funding:** This work was supported by funding from the University of Washington Diabetes

## Abstract

Estrogens are important for maintaining metabolic health in males. However, the key sources of local estrogen production for regulating energy metabolism have not been fully defined. Immune cells exhibit aromatase activity and are resident in metabolic tissues. To determine the relative contribution of immune cell-derived estrogens for metabolic health in males, C57BL6/J mice underwent bone marrow transplant with marrow from either wild-type (WT(WT)) or aromatase-deficient (WT(ArKO)) donors. Body weight, body composition, and glucose and insulin tolerance were assessed over 24 weeks with mice maintained on a regular chow diet. No differences were found in insulin sensitivity between groups, but WT(ArKO) mice were more glucose tolerant than WT(WT) mice 20 weeks after transplant, suggestive of enhanced glucose disposal (AUC$_{glucose}$ 6061±3349 in WT(WT) mice versus 3406±1367 in WT(ArKO) mice, p = 0.01). Consistent with this, skeletal muscle from WT(ArKO) mice showed higher expression of the mitochondrial genes *Ppargc1a* (p = 0.03) and *Nrf1* (p = 0.01), as well as glucose transporter type 4 (GLUT4, *Scl2a4*; p = 0.02). Skeletal muscle from WT(ArKO) mice had a lower concentration of 17β-estradiol (5489±2189 pg/gm in WT(WT) mice versus 3836±2160 pg/gm in WT(ArKO) mice, p = 0.08) but higher expression of estrogen receptor-α (ERα, *Esr1*), raising the possibility that aromatase deficiency in immune cells led to a compensatory increase in ERα signaling. No differences between groups were found with regard to body weight, adiposity, or gene expression within adipose tissue or liver. Immune cells are a key source of local 17β-estradiol production and contribute to metabolic regulation in males, particularly within skeletal muscle. The respective intracrine and paracrine roles of immune cell-derived estrogens require further delineation, as do the pathways that regulate aromatase activity in immune cells specifically within metabolic tissues.

Research Center grant P30 DK01704 (KBR, http://depts.washington.edu/diabetes/), the American Heart Association grant 16GRNT30700006 (KBR, www.heart.org) and the NIH National Center for Complimentary and Integrative Health grant K01 AT007177 (LJdH, https://nccih.nih.gov/). The funders had no role in study design, data collection, analysis, decision to publish, or preparation of the manuscript.

**Competing interests:** The authors have declared that no competing interests exist.

# Introduction

Estrogens are now recognized to play important metabolic roles in men. Initial evidence that men require adequate estrogen exposure for metabolic health derived from men with rare genetic syndromes of estrogen deficiency. Thus, men with loss of either functional estrogen receptor-α (ERα) or aromatase—the enzyme that generates estrogens from androgen precursors–exhibit metabolic derangements including increased visceral adiposity, insulin resistance, and reduced bone mineral density [1, 2]. More recently, clinical intervention studies have shown that short-term estrogen deprivation in men results in increased adiposity [3, 4] and reduced insulin sensitivity [5]. In parallel with these clinical observations, genetic mouse models also have demonstrated the importance of estrogens for regulating energy balance, body composition, insulin sensitivity, and skeletal health in males [6–8]. Similar to men with estrogen insufficiency, male mice with global aromatase deficiency exhibit increased adiposity, insulin resistance, and lower bone mass [9, 10]. However, the mechanisms underlying estrogen-mediated metabolic regulation in males remain incompletely understood, and the key tissue-specific sites of estrogen action, as well as local sources of estrogen production, have to yet be clearly defined.

Estrogens are generated not only from classically steroidogenic tissues including gonads and adrenal glands but also are generated locally within tissues, and numerous cell types are able to convert androgens to estrogens through aromatase activity [11, 12]. Thus, tissue levels of estrogens, including the most important bioactive estrogen, 17β-estradiol, may be dissociated from circulating levels [13, 14]. Adipose tissue stromal cells express aromatase and are a well-established source of both local and circulating estrogens [11]. Aromatase expression and activity similarly have been demonstrated in hepatocytes, myocytes, and osteoblasts, underscoring the importance of local estrogen production in key metabolic tissues. Notably, immune cells are present in all key metabolic tissues, and aromatase activity has been identified in immune cells including macrophages and both T and B lymphocytes [15–17]. Therefore, the possibility exists that aromatase activity within immune cells generates bioactive estrogens that mediate cell-intrinsic, intracrine effects, with consequent changes in cellular phenotype and function. Further, immune cells may secrete estrogens and could thereby contribute to tissue-specific estrogen concentrations with signaling effects conferred in neighboring cells.

No studies to date have examined the metabolic effects of estrogens generated specifically within immune cells. We used a bone marrow transplant model to reconstitute hematopoietic cells in wild-type male mice with either wild-type (WT) or aromatase deficient (ArKO) hematopoietic cells. Our goal was to determine the metabolic effects of aromatase deficiency in hematopoietic cells in male mice. We predicted that loss of aromatase function selectively within hematopoietic cells would lead to increased adiposity and impaired glucose tolerance, reproducing the metabolic phenotype of male mice with global aromatase deficiency.

# Materials and methods

## Measurement of immune cell estrogen secretion

To isolate peritoneal immune cells, thioglycolate (BD; Franklin Lakes, NJ) was administered to male C57Bl6/J mice by intra-peritoneal injection as described previously [18]. Five days subsequent to injection, peritoneal fluid was collected by lavage. Cells were separated by centrifugation of lavage fluid and plated at a density of $2x10^6$ per well in 6-well tissue culture plates. Macrophages were enriched through adhesion purification and cultured 24 hours in RPMI media with 10% fetal bovine serum and 1% penicillin/streptomycin.

Immune cell conditioned media was collected, and immune cell estrogen secretion was quantified by liquid chromatography-tandem mass spectrometry (LC-MS/MS) as others have

described previously [19]. In duplicate, 0.1 mL of media from each sample was transferred to glass tubes containing 25 μL of internal standard ([5]d-estradiol, [4]d -estrone, 4pg/ul) and vortexed. Samples were held at room temperature for 10 minutes, followed by addition of 2.5 mL of hexane: ethyl Acetate (8:2 v/v). Samples were rotated for 15 min, then spun at 4500 rpm in the microcentrifuge for 15 min at 4˚C. The upper solvent phase (2 mL) was transferred to a clean glass tube, and this procedure was repeated using 1.5 mL of hexane: ethyl Acetate (8:2 v/v) Hexane: ethyl acetate was then removed using a Speed-Vac with solvent trap while heating to 38˚C for 30 minutes. Next, 0.3ml of hexane: ethyl Acetate (8:2 v/v) was added to sample tube and vortexed, then removed again by Speed-Vac. Sodium bicarbonate (25 μL of 100 mM, pH 10.5) and 25 μl of dansyl chloride in acetone (1 g/L) were added to sample vial, and samples were spun briefly at 1000 rpm. Finally, samples were heated to 60˚C for 3 min and again briefly spun at 1000 rpm. Samples were transferred to a clean glass conical HPLC vial. The vial was then placed in 1.8ml conical MF tube and spun at 13,000 rpm for 5 minutes in microcentrifuge. Without disrupting pellet, 40 μl of sample was transferred to a clean conical HPLC vial. Samples were stored overnight at room temperature and then at 4˚C. Extraction and derivatization of the standards and samples were performed simultaneously. Samples were run on an AB Sciex 5500 QTRAP tandem quadrupole mass spectrometer. Chromatographic separation performed using a Phenomenex Kinetex 1.7u 100 x 2.1 mm Phenyl-Hexyl column. The inter-assay CVs for estrone and estradiol were 12.1% and 5.6%, respectively, with a lower limit of detection of 3.6 pmol/L for both.

## Animals and study design

Generation of donor mice: Male and female mice heterozygous for aromatase deficiency were generated as previously described [8, 9] and bred to generate littermate wild-type (WT) and aromatase-deficient (ArKO) male mice. The breeder mice were from a congenic C57BL6/J strain and provided from an animal colony at the University of Texas-Southwestern maintained by the senior author (OKO); these included 1 male and 2 female mice heterozygous for aromatase deficiency and 1 wild-type male mouse. The offspring served as bone marrow donors (n = 3 ArKO donor male mice and n = 4 WT donor male mice). For animal genotyping, tissue harvested through tail clip was boiled in NaOH (50 Mm, 600 μL) for 10–20 minutes then vortexed for 5–10 sec and neutralized with 1M Tris (100 μL, pH 8.0). Samples were centrifuged for 6 min at 12,000 rpm to pellet debris, and 200 μL was transferred to a fresh tube. For each sample, 1–2 μL was used for PCR in a 25 μL reaction. PCR was performed as detailed elsewhere [20] using Taq DNA polymerase (Thermo Fisher Scientific; Waltham, MA) and Redi Load gel loading buffer (Invitrogen; Carlsbad, CA). The following primer pairs were used: Neo F: 5' ATC AGG ATG ATC TGG ACG AAG 3,' Neo R: 5' CCA CAG TCG ATG AAT CCA GAA 3,' exon 9 F: 5' GTG ACA GAG ACA TAA AGA TCG 3,' and exon 9 R: 5' GTA AAT TCA TTG GGC TTA GGG 3.' PCR products were run on a 2.5% agarose gel, with product sizes of 170 bp for Neo and 220 bp for exon 9.

Bone marrow transplant procedure: Wild-type C57BL6/J male mice (n = 30) were purchased through the Jackson Laboratory (Bar Harbor, ME; strain #000664) and served as bone marrow transplant recipients. Mice were allowed to acclimate after transfer for 4 weeks and underwent transplant at 9 weeks of age (Fig 1A). Recipient mice were subject to lethal irradiation (10 Gy) and administered neomycin (2 mg/mL) in drinking water for 2 weeks post-irradiation. Recipients of bone marrow transplant were monitored 3–4 times each week for 4 weeks subsequent to irradiation until marrow engraftment occurred. Mice were monitored for any signs or behaviors indicative of poor health, including hunching, difficulty breathing, or listlessness, and any concerning signs or behaviors were reported to veterinary medicine.

## A. Schematic of the study design

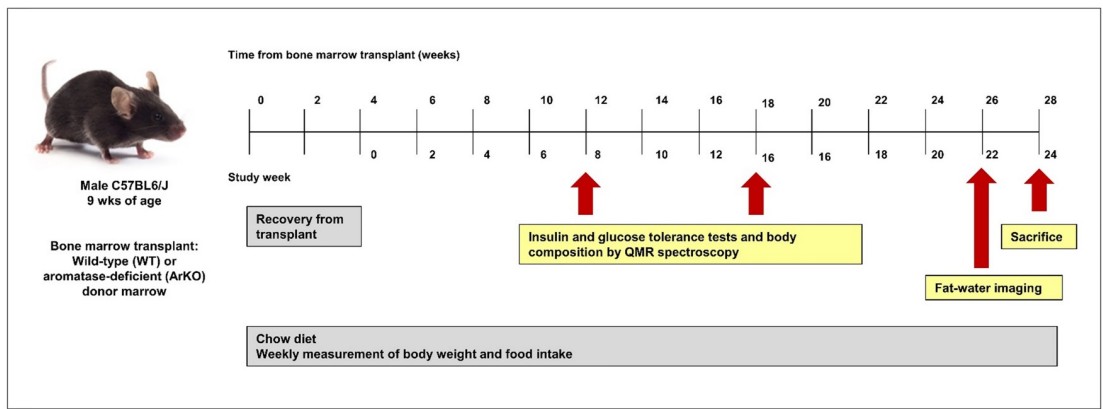

## B. Secretion of estrogens by peritoneal immune cells

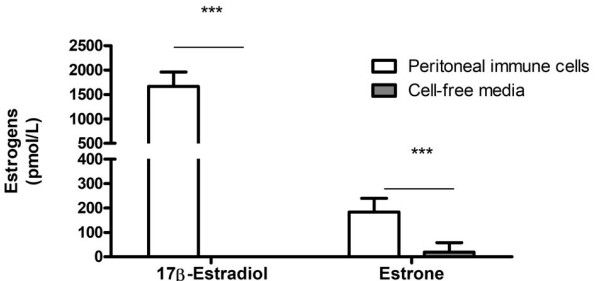

**Fig 1. Schematic of the study design and measurement of immune cell secretion of estrogens.** A metabolic phenotyping study was performed in wild-type male mice transplanted with bone marrow from wild-type or aromatase-deficient donors (A). Concentrations of 17β-estradiol and estrone were markedly higher in media conditioned by murine peritoneal immune cells than cell-free media (n = 8 samples conditioned media, n = 4 samples cell-free media; ***$p<0.0001$) (B).

The morning following irradiation, donor mice were sacrificed by $CO_2$ inhalation and exsanguination, and femurs and tibias were harvested. Femurs and tibias were washed in ethanol once followed by 3 washes in phosphate-buffered saline (PBS). A single end of the bone was clipped, and bones were placed in 600 μL tubes punctured with an 18" needle and placed within a 1.5 mL tube. The bones were then centrifuged at 10$g$ for 8 seconds, and bone marrow was collected in the 1.5 mL tube. Bone marrow from each tube was immediately suspended in 1 mL red blood cell lysis buffer (Sigma-Aldrich; St. Louis, MO), and marrow from donor mice of each genotype was then pooled in 50 mL tubes. To quench the lysis reaction, 3–5 mL PBS was added to each tube, and samples were centrifuged at 400$g$ for 5 minutes. The supernatant was aspirated, and the cell pellet was resuspended in 10 mL PBS. Cells were counted using a hemocytometer, and cells were resuspended in 1% PBS to a final concentration of 23.8x10$^6$ cells/mL. Pooled cells were aliquoted into syringes for bone marrow transplant. A total of 7x10$^6$ cells (300 μL injection volume) was administered to irradiated recipient mice through retro-orbital injection on the day following irradiation. Thus, all bone marrow transplant recipients were WT mice with marrow from either WT [WT(WT)] or ArKO [WT(ArKO)] donors.

During the post-transplant period, 3 mice [2 WT(WT), 1 WT(ArKO)] died, presumably due to infection prior to engraftment of transplanted cells. Another mouse [(WT(ArKO)] died

unexpectedly during an insulin tolerance test, presumably due to hypoglycemia. Consequently, a total of 26 mice (n = 13 per group) remained and were included in the phenotyping study. During fat-water imaging for detailed body fat quantification, 2 animals in the WT(WT) group died during recovery from anesthesia. Therefore, for all post-mortem tissue analyses, a total of 24 mice were included (n = 11 WT(WT) animals and n = 13 WT(ArKO) animals). Bone marrow engraftment was verified through genotyping of circulating immune cells. At 8 weeks after transplant (study week 4), whole blood was collected and centrifuged for plasma preparation, and the supernatant was removed. NaOH (600 μL) was added to the remaining cell pellet, and samples were then placed on a heat block for 20 minutes. Genotyping was performed on 1 μL of the end product.

Body weight and food intake were measured weekly. Glucose and insulin tolerance tests were performed at study weeks 8 and 16 (12 and 20 weeks after bone marrow transplant, respectively). Body composition was measured by QMR spectroscopy at study weeks 8 and 16. Measurement of visceral, subcutaneous, and liver fat volume subsequently was performed using fat-water imaging (study week 22), and animals were sacrificed at study week 24 through cervical dislocation and exsanguination.

For the duration of the study, mice were maintained on a regular chow diet (D12450H, Research Diets, Inc; New Brunswick, NJ, USA). All mice were group housed (4–5 animals per cage, breeding pairs 2 animals per cage) with ad libitum access to food and water. In order to minimize suffering and risk to animals, fasting times were limited to 4–5 hours prior to metabolic testing, bone marrow transplant and sacrifice were performed under isoflurane anesthesia, and mice were placed on heating pads immediately following fat-water imaging. Adequacy of anesthesia was assessed through monitoring of respiratory rate and toe pinch response. All study procedures were approved by the University of Washington Institutional Animal Care and Use Committee (IACUC, protocol #4369–01).

### Immunohistochemistry and adipocyte size measurement

Antigen presenting cells in formalin-fixed, paraffin-embedded epididymal and inguinal adipose tissue were quantified using a rat monoclonal antibody against MAC2 (1:2500, Cedarlane Laboratories), as described previously [21]. Adipocyte number and cross-sectional area were measured using computer image analysis with Image J software as previously described [22].

### Intra-peritoneal glucose and insulin tolerance tests

At study weeks 8 and 16, glucose homeostasis was assessed through glucose (GTT) and insulin (ITT) tolerance tests. Animals were fasted for 5 hours prior to testing and administered insulin (0.75 U/kg body weight) or glucose (1.5 g/kg body weight, 20% dextrose solution) by intra-peritoneal injection. Blood glucose was measured at baseline and 15, 30, 60, 90, and 120 minutes after injection using a hand-held glucose meter (Accu-Chek; Roche Diabetes Care, Inc; Basel, SUI).

### Body composition assessment

Body composition was assessed at study weeks 8 and 16 by QMR spectroscopy performed through the University of Washington Nutrition Obesity Research Center (UW NORC) Energy Balance Core [22, 23]. At study week 22, body composition also was measured by fat-water imaging using a 3-Tesla magnetic resonance system to assess body fat distribution. This method was used to quantify total, visceral and subcutaneous fat volumes as well as liver fat content and has been previously described in detail [22]. All fat-water imaging analyses were performed by a single radiologist blinded to animal grouping.

**Table 1. Primer sets employed.**

| Gene name | Thermo Fisher Scientific accession number | Gene name | Thermo Fisher Scientific accession number |
|---|---|---|---|
| *Acaca* | Mm01304257_m1 | *Lep* | Mm00434759_m1 |
| *Actb* | Mm02619580_g1 | *Lpl* | Mm00434770_m1 |
| *Adipoq* | Mm00456425_m1 | *Nono* | Mm00834875_g1 |
| *Atgl* | Mm00503040_m1 | *Nrf1* | Mm01135606_m1 |
| *B2m* | Mm00437762_m1 | *Pck1* | Mm01184322_m1 |
| *Ccl2* | Mm00441242_m1 | *Pgk1* | Mm00435617_m1 |
| *Cebpa* | Mm00514283_s1 | *Ppargc1a* | Mm00447184_g1 |
| *Cebpg* | Mm01266786_m1 | *Ppara* | Mm00440939_m1 |
| *Cpt1b* | Mm00487200_m1 | *Pparg* | Mm01184322_m1 |
| *Cyp19a1* | Mm00484049_m1 | *Scd1* | Mm00772290_m1 |
| *Dgat1* | Mm00515643_m1 | *Slc2a4* | Mm00436615_m1 |
| *Dgat2* | Mm00499536_m1 | *Srebf1* | Mm01163722_g1 |
| *Dlk1* | Mm00494477_m1 | *Tfam* | Mm00447485_m1 |
| *Emr1* | Mm00802530_m1 | *Tgfb1* | Mm01178829_m1 |
| *Esr1* | Mm00433149_m1 | *Tnf* | Mm00443258_m1 |
| *Hsl* | Mm00495359_m1 | *Ucp1* | Mm01244860_m1 |
| *Il6* | Mm00446190_m1 | *Ucp2* | Mm00627599_m1 |
| *Irs1* | Mm01278327_m1 | | |

### Metabolic tissue gene expression analysis

At sacrifice, liver, skeletal muscle, and inguinal and epididymal adipose tissue were harvested, flash frozen, and RNA was extracted from ~100 mg of each tissue type for each animal using RNeasy mini-kits (Qiagen, Germantown, Maryland, USA). Generation of cDNA and quantitative real-time PCR were performed using Taqman probe sets (Thermo Fisher Scientific, Waltham, MA, USA) as previously described [22]. Gene expression data were analyzed using the ΔΔCt method. Expression of 3 candidate housekeeping genes was measured for each tissue, and the genes exhibiting the most stable tissue-specific expression were utilized for data normalization. Gene expression was normalized to β-2-microglobulin (*B2m*) for liver, the geometric mean of *B2m* and non-POU-containing domain, octamer binding protein (*NoNo*) for adipose tissue, and β-actin (*Actb*) for skeletal muscle. See Table 1 for a complete list of Taqman primer sets used.

### Plasma analyses and measurement of hepatic lipids

Plasma insulin, adiponectin, leptin, and IL-6 concentrations were measured at study weeks 8 and 24 (sacrifice) through commercially available ELISA kits (Millipore; Billerica, MA) according to the manufacturer's instructions. Plasma 17β-estradiol concentrations were measured at week 24 (sacrifice) through ELISA (Calbiotech, Inc; El Cajon, CA). A modified Folch method was used to extract hepatic lipids [24], and colorimetric assays were used to measure hepatic cholesterol and triglyceride levels (Stanbio Laboratory; Boerne, TX) [22].

### Quantification of 17β-estradiol and estrone concentrations within metabolic tissues

Tissue concentrations of estrone and 17β-estradiol were measured for skeletal muscle, liver, and epididymal adipose tissue. Inadequate sample remained for inguinal adipose tissue for

these analyses. Estrogen concentrations were quantified through ultra-performance liquid chromatography tandem mass spectrometry (UPLC/MS-MS) as previously described in detail [25, 26]. The current analyses differed from prior methods only in that a Quatro-micro triple-quad mass spectrometer (Waters Corporation; Milton, MA) was employed. All other analytical parameters were similar to prior methods.

### Statistical analyses

Unpaired Student's t test was used to evaluate between-group differences at a single timepoint. Repeated measures ANOVA (RM-ANOVA) was used to determine whether an interaction between time and group was evident for study endpoints measured at more than 1 timepoint. Data are presented as mean ± standard deviation (SD) for cell culture data and mean ± standard error of the mean (SEM) for the remaining data. A p-value threshold of $p<0.05$ was used for statistical significance, and GraphPad Prism 5.0 software (GraphPad Software, Inc; La Jolla, CA) was used for figure creation and all statistical analyses.

## Results

### Peritoneal immune cells from male mice secrete estrogens

LC-MS/MS was used to quantify estrogen secretion from peritoneal immune cells to better define patterns of estrogen production. Thioglycolate-elicited peritoneal immune cells were harvested from male mice and cultured overnight. Steroid measurement demonstrated marked generation of 17β-estradiol (1667±295 pmol/L in conditioned media vs. <37 pmol/L in cell-free media; $p<0.0001$) and lesser production of estrone (196±28 pmol/L in conditioned media vs. 18±41 pmol/L in cell-free media; $p<0.0001$) (Fig 1B).

### Aromatase deficiency in bone marrow-derived cells does not alter body weight or body composition in male mice

To verify successful engraftment of transplanted bone marrow, whole blood was collected 8 weeks post-transplant, and genotyping was performed on circulating white blood cells (Fig 2A). No differences in absolute or change in body weight (Fig 2B and 2C) were evident between WT(WT) and WT(ArKO) mice over 24 weeks on a regular chow diet, and body weights in both groups were nearly identical at sacrifice (26.4±0.5 gm in WT(WT) mice versus 26.5±0.5 gm in WT(ArKO) mice, p = 0.91).

At study week 8 (12 weeks after bone marrow transplant), WT(ArKO) mice had significantly greater total fat mass than WT(WT) mice (1.7±0.1 gm versus 1.4±0.1 gm, respectively, p = 0.04; Fig 3A), but this difference in fat mass was no longer evident at study week 16 (Fig 3B). No differences in lean mass were seen between groups at either timepoint. Adiposity was measured again through fat-water imaging at study week 22 to assess body fat distribution. No differences in subcutaneous, visceral, or total fat volume were found between groups (Fig 3C). At sacrifice, epididymal adipose tissue mass did not differ significantly between WT(WT) and WT(ArKO) mice (18±1 mg/gm body weight in WT(ArKO) mice versus 16±1 mg/gm body weight in WT(WT) mice, p = 0.09; Fig 3D), and no difference was found between groups in inguinal white adipose tissue mass.

Liver weights were comparable between both groups (50±1 mg/gm body weight in WT (WT) mice versus 48±2 mg/gm body weight in WT(ArKO) mice, p = 0.50; Fig 3D). Further, liver triglyceride and cholesterol content did not differ between WT(WT) and WT(ArKO) mice (Fig 3E), nor did total liver fat volume differ between groups as quantified by fat-water imaging (Fig 3F).

**A. Verification of donor bone marrow engraftment**

**B. Body weight**

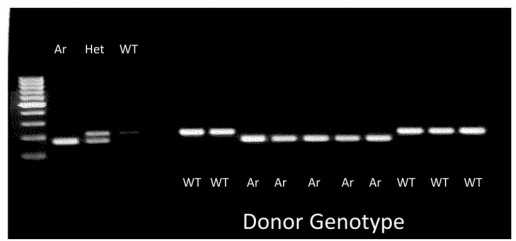

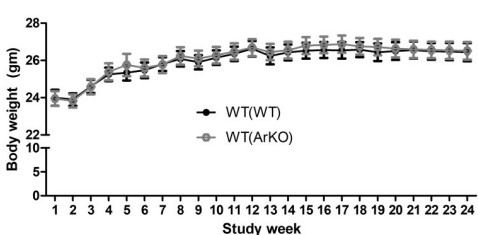

**C. Body weight change**

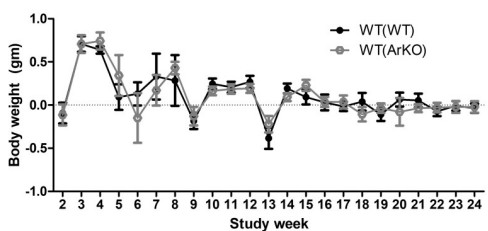

**Fig 2. Verification of bone marrow engraftment and body weight in wild-type (WT) male mice transplanted with bone marrow from either WT (WT(WT)) or aromatase-deficient (WT(ArKO)) donors.** Genotyping of circulating white blood cells was performed to verify successful bone marrow engraftment (A), with illustrative results from 10 animals shown. No differences between groups were found in either absolute body weight (B) or change in body weight (C) over the 24-week study.

**A. Body composition at week 8**

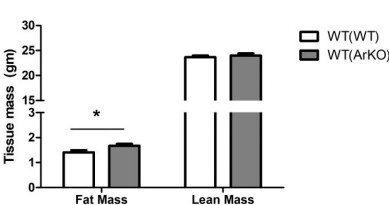

**B. Body composition at week 16**

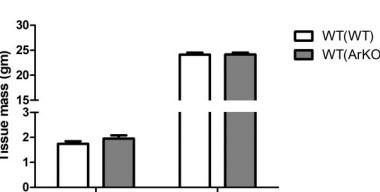

**C. Body fat distribution**

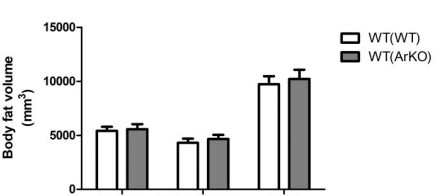

**D. Tissue weight**

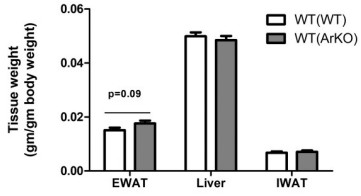

**E. Liver fat content**

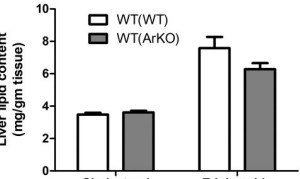

**F. Liver fat volume**

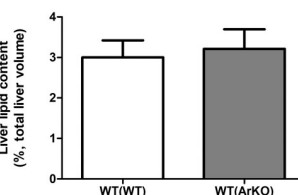

**Fig 3. Body composition in wild-type (WT) male mice transplanted with bone marrow from either WT (WT(WT)) or aromatase-deficient (WT(ArKO)) donors.** WT(ArKO) mice exhibited greater total body fat mass at week 8 (A) but not week 16 (B). At week 22, fat mass distribution was similar between groups (C). At sacrifice, tissue weights were similar between groups, although a trend was found toward greater epididymal fat pad weight in WT(ArKO) mice relative to WT(WT) mice (D). Liver lipid content was comparable between groups whether quantified by cholesterol and triglyceride mass (E) or liver fat volume as measured by MRI (F). *p<0.05.

## Male mice with hematopoietic aromatase deficiency exhibit enhanced glucose tolerance

Glucose tolerance and insulin sensitivity were assessed in WT(WT) and WT(ArKO) mice at study weeks 8 and 16 (12 and 20 weeks after bone marrow transplant, respectively). At study week 8, blood glucose did not differ between groups at any single timepoint during the glucose tolerance test, nor was the glucose area under the curve (AUC) different between groups ($AUC_{glucose}$ 3685±543 in WT(WT) mice versus 4825±684 in WT(ArKO) mice, p = 0.23; Fig 4A). Similarly, insulin sensitivity was comparable between groups at study week 8 (Fig 4B). However, at study week 16, WT(ArKO) mice exhibited significantly lower excursions in blood glucose during the glucose tolerance test (Fig 4C), as well as a lower glucose AUC ($AUC_{glucose}$ 6061±929 in WT(WT) mice versus 3406±379 in WT(ArKO) mice, p = 0.01), indicative of enhanced glucose tolerance compared to WT(WT) mice. Again, no difference in insulin sensitivity was seen between groups at study week 16 (Fig 4D).

Consistent with findings of comparable insulin sensitivity between WT(WT) and WT(ArKO) mice, fasting plasma concentrations of insulin and glucose were similar between groups at study week 8 and at sacrifice (study week 24, 28 weeks after bone marrow transplant; Fig 5A and 5B). No differences were found between groups in plasma concentrations of the adipokines leptin (Fig 5C) or adiponectin (Fig 5D), nor did plasma lipid concentrations differ between WT(WT) and WT(ArKO) mice (Fig 5E and 5F). Finally, plasma concentrations of 17β-estradiol were similar between groups at baseline, study week 8, and sacrifice (Fig 5G).

## Loss of aromatase in hematopoietic cells confers increased skeletal muscle expression of genes implicated in insulin signaling and mitochondrial function

Consistent with the differences found in systemic glucose tolerance between WT(WT) and WT(ArKO) mice, differences were found between groups in the expression of genes

**Fig 4. Glucose tolerance and insulin sensitivity in wild-type (WT) male mice transplanted with bone marrow from either WT (WT(WT)) or aromatase-deficient (WT(ArKO)) donors.** At study week 8, no differences were seen in either glucose tolerance (A) or insulin sensitivity (B) between groups. In contrast, at study week 16, glucose tolerance was enhanced in WT(ArKO) mice relative to WT(WT) mice (C). Insulin sensitivity remained comparable between groups (D).

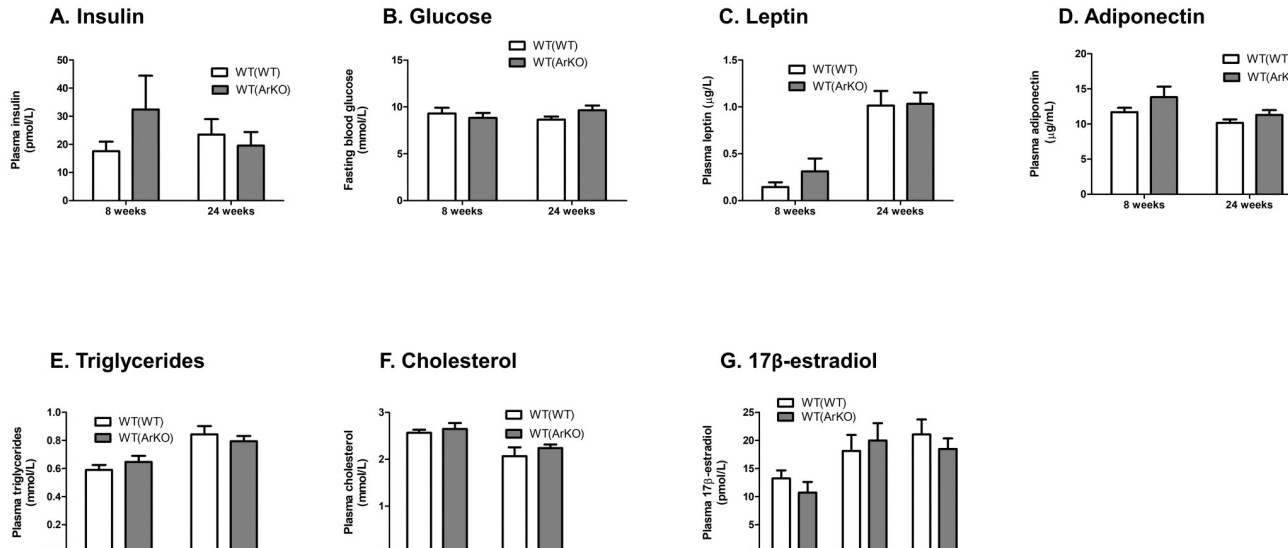

**Fig 5. Plasma analytes in wild-type (WT) male mice transplanted with bone marrow from either WT (WT(WT)) or aromatase-deficient (WT (ArKO)) donors.** At both study week 8 and study week 24, no differences were found between groups in fasting plasma concentrations of insulin (A), glucose (B), leptin (C), or adiponectin (D). Fasting plasma cholesterol (E) and triglyceride (F) concentrations were similar between groups at both timepoints, and plasma concentrations of 17β-estradiol remained comparable between groups throughout the study (G).

implicated in energy metabolism within skeletal muscle (Fig 6A). Skeletal muscle from WT (ArKO) mice exhibited higher mRNA expression of glucose transporter type 4 (GLUT4, *Slc2a4*; p = 0.02), consistent with the finding of enhanced systemic glucose disposal. Further, WT(ArKO) mice had higher mRNA expression of the mitochondrial genes peroxisome proliferator-activated receptor gamma coactivator 1-alpha (PGC-1α, *Ppargc1a*; p = 0.03) and nuclear transcription factor 1 (NRF1, *Nrf1*; p = 0.01), suggesting enhanced energy metabolism and mitochondrial biogenesis. Skeletal muscle from WT(ArKO) also showed lower expression of the lipogenic enzyme stearoyl CoA desaturase-1 (SCD1, *Scd1*; p = 0.03), suggesting reduced ectopic lipid deposition. In WT(ArKO) mice, mean expression of ERα (*Esr1*) was higher relative to WT(WT) mice, although this difference did not achieve statistical significance (p = 0.09). Collectively these results suggest improvement in glucose and lipid metabolism in skeletal muscle from mice with hematopoietic aromatase deficiency.

In contrast, no differences in mRNA expression were found for any of the targeted genes in liver (Fig 6B), epididymal adipose tissue (Fig 6C) or inguinal adipose tissue (Fig 6D). Whereas a trend was seen toward higher ERα (*Esr1*) expression among WT(ArKO) mice in skeletal muscle, no differences in ERα mRNA expression were found between WT(ArKO) and WT (WT) mice in liver, epididymal adipose tissue, or inguinal adipose tissue. Among these metabolic tissues, aromatase (*Cyp19a*) expression was detectable only in epididymal adipose tissue, and expression levels did not differ between groups.

## Adipose tissue estrogen concentrations and immune cell infiltration

In epididymal adipose tissue, both estrone and 17β-estradiol concentrations were similar between groups (Fig 7A). Similarly, in liver, no between-group differences were found in tissue estrone concentration (485±165 pg/gm in WT(WT) mice versus 257±59 pg/gm in WT(ArKO) mice, p = 0.18) or 17β-estradiol concentration (3205±546 pg/mL in WT(WT)

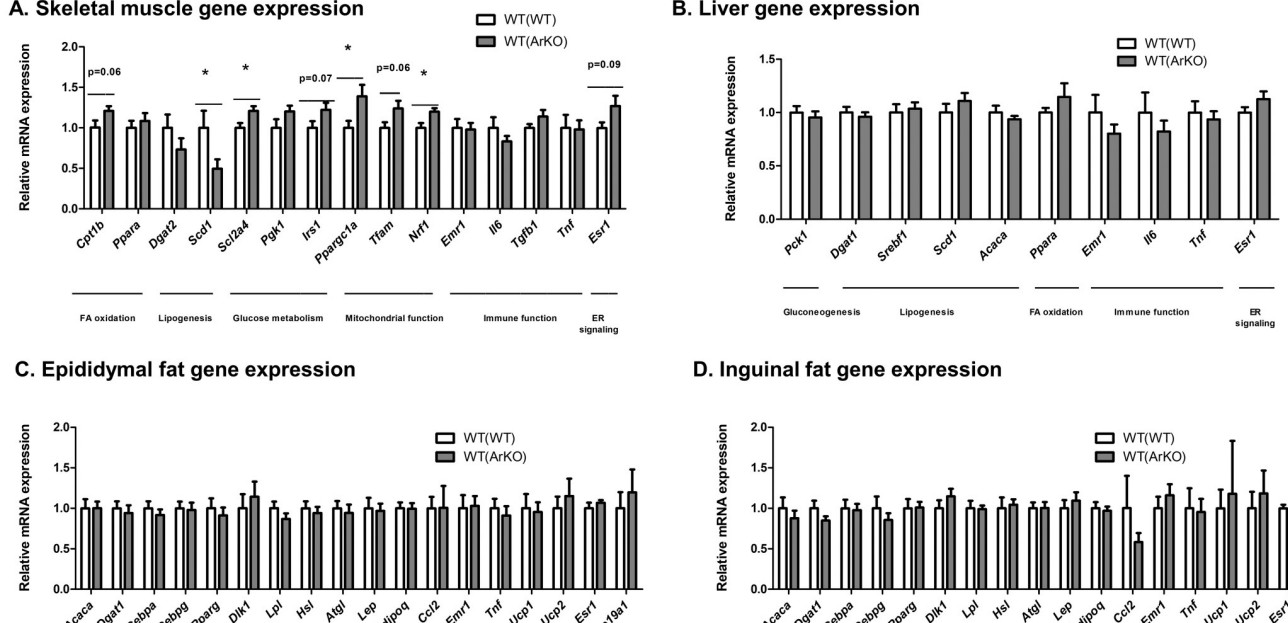

**Fig 6. Gene expression in metabolic tissues from wild-type (WT) male mice transplanted with bone marrow from either WT (WT(WT)) or aromatase-deficient (WT(ArKO)) donors.** Marked differences in skeletal muscle mRNA expression were found between groups (A), but gene expression profiles in liver (B), epididymal adipose tissue (C), and inguinal adipose tissue (D) were comparable in WT(ArKO) and WT(WT) mice. *p<0.05.

mice versus 2338±199 pg/mL in WT(ArKO) mice, p = 0.13; Fig 7B). In skeletal muscle, tissue estrone concentrations were comparable between groups (5804±967 pg/gm in WT (WT) mice versus 4336±392 pg/gm in WT(ArKO) mice, p = 0.15), but tissue 17β-estradiol concentrations were ~30% lower in WT(ArKO) mice relative to WT(WT) controls, although the difference did not achieve significance (5489±660 pg/gm in WT(WT) mice versus 3836±599 pg/gm in WT(ArKO) mice, p = 0.08; Fig 7C). Notably, this lack of significance was driven by a single outlier in the WT(ArKO) group, with a value >2 standard deviations above the mean for the remainder of the group. When this mouse was excluded from the analysis, tissue 17β-estradiol concentrations were significantly different between groups (5489±660 pg/gm in WT(WT) mice versus 3507±544 pg/gm in WT(ArKO) mice, p = 0.03).

Within adipose tissue, no differences between WT(WT) and WT(ArKO) mice were observed in the number of infiltrating antigen-presenting cells as quantified by positive Mac-2 staining. Thus, in both epididymal (Fig 8A) and inguinal (Fig 8B) adipose tissue, comparable numbers of antigen-presenting cells were found between groups whether quantified as a percentage of total area or as cell number per field. Adipocyte size did not differ between groups in either inguinal or epididymal adipose tissue (Fig 8C and 8D).

## Discussion

These findings demonstrate novel evidence that immune cells contribute to total estrogen generation in metabolic tissues and support the importance of immune cell-derived estrogens for regulating metabolic health in males. Surprisingly, given the protective roles played by 17β-

**A. Epididymal adipose tissue estrogens**

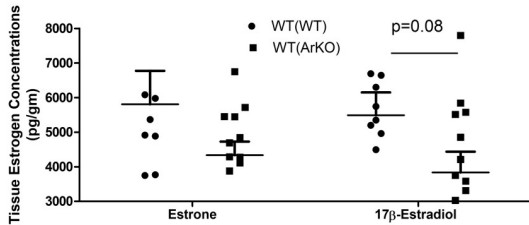

**B. Liver estrogens**

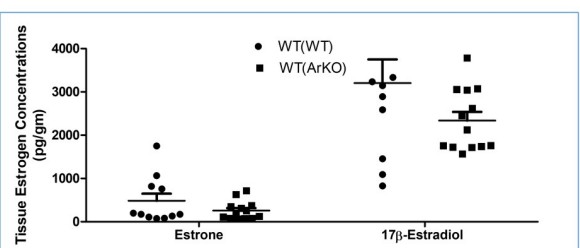

**C. Skeletal muscle estrogens**

**Fig 7. Tissue estrogen concentrations in wild-type (WT) male mice transplanted with bone marrow from either WT (WT(WT)) or aromatase-deficient (WT(ArKO)) donors.** Tissue concentrations of estrone and 17β-estradiol were similar in epididymal adipose tissue (A) and liver (B). However, in skeletal muscle, a trend toward a lower concentration of 17β-estradiol was evident in WT(ArKO) mice relative to WT(WT) mice (C). Some values appear missing due to overlapping points in the scatterplots.

estradiol in tissue glucose homeostasis, male mice with selective aromatase deficiency in hematopoietic cells developed improved glucose tolerance relative to controls by 20 weeks after bone marrow transplant. This enhanced glucose tolerance occurred in the absence of differences in body weight, lean body mass, or adiposity between WT(WT) and WT(ArKO) mice. In parallel with enhanced systemic glucose tolerance, WT(ArKO) mice exhibited differential gene expression within skeletal muscle remarkable for greater expression of genes implicated in insulin signaling and mitochondrial function. Importantly, despite a lower tissue concentration of 17β-estradiol, WT(ArKO) mice also exhibited increased mRNA expression of ERα (*Esr1*) in skeletal muscle, indicating that loss of aromatase activity in hematopoietic cells may have led to a paradoxical increase in tissue-specific ERα signaling. Consistent with this idea, skeletal muscle exhibited enhanced expression of genes known to be positively regulated by 17β-estradiol. Interestingly, this phenomenon was present only in skeletal muscle, whereas adipose tissue and liver showed comparable estrogen concentrations and ERα mRNA expression in both groups. Importantly, however, these results are specific to a regular chow diet and may vary considerably in the setting of obesity, high-fat feeding, or other metabolic stressors. Collectively, these results implicate aromatase activity in immune cells particularly in skeletal muscle energy metabolism and underscore the need to better understand the intracrine and paracrine mechanisms through which immune cell-derived estrogens contribute to metabolic regulation in males.

Our study demonstrated reduced skeletal muscle 17β-estradiol concentrations but enhanced glucose tolerance in male mice with hematopoietic aromatase deficiency. This finding is in clear contrast to prior studies demonstrating impaired glucose tolerance in male mice with global aromatase deficiency or in female mice with loss of estrogen signaling in

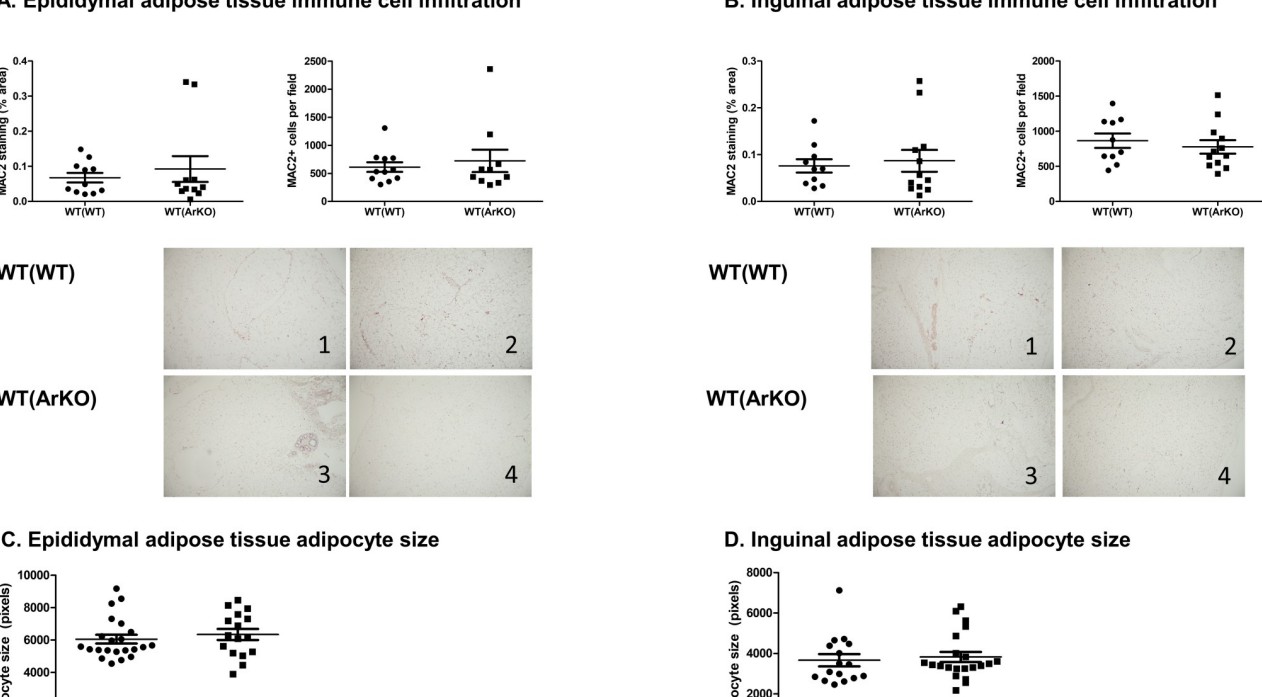

**Fig 8. Adipose tissue histology in wild-type (WT) male mice transplanted with bone marrow from either WT (WT(WT)) or aromatase-deficient (WT(ArKO)) donors.** Infiltration of antigen-presenting cells was similar in WT(WT) and WT(ArKO) mice in both epididymal (A) and inguinal (B) fat. Below graphs are representative images of Mac-2 staining of epididymal and inguinal adipose tissue from 2 animals in each group. Adipocyte size was similar in WT(WT) and WT(ArKO) mice in both epididymal (C) and inguinal (D) adipose tissue depots.

skeletal muscle conferred by ERα deficiency in either myocytes or myeloid cells [27–29]. Notably, however, tissue analyses also demonstrated higher ERα mRNA expression in skeletal muscle from WT(ArKO) mice, suggesting the possibility that selective loss of aromatase activity in immune cells yielded a compensatory increase in total estradiol signaling within skeletal muscle. A compensatory increase in estradiol signaling is further supported by the fact that skeletal muscle from WT(ArKO) mice showed enhanced expression of genes that are specifically upregulated by 17β-estradiol, including those involved in mitochondrial biogenesis [30]. Further, systemic administration of 17β-estradiol to orchiectomized mice enhanced skeletal muscle glucose uptake due to upregulation of GLUT4 expression [31], another gene that exhibited increased expression in the WT(ArKO) mice relative to WT(WT) controls. An alternative explanation for our findings is that the tissue-specific metabolic effects of estrogens are contingent on the cell of origin; thus, immune cell-derived estrogens could inhibit insulin signaling and mitochondrial biogenesis, and the protective metabolic effects of 17β-estradiol may derive exclusively from aromatase activity outside of the hematopoietic compartment.

Immune cells are resident in skeletal muscle, although they constitute well under 1% of total cells in muscle tissue [32]; these findings therefore are remarkable in that they suggest that immune cells are a key source of tissue estrogen generation and, despite their small numbers, contribute substantially to both total tissue estradiol concentrations and estradiol-mediated regulation of insulin sensitivity and mitochondrial function in skeletal muscle. Notably, although tissue estradiol concentrations differed by roughly 1/3, this finding did not achieve

statistical significance and requires verification in a future study powered for this endpoint. As macrophages and lymphocytes are both present within skeletal muscle, additional work is needed to determine the respective contribution of each of these cell types to the observed differences in tissue estradiol concentrations, tissue-specific energy metabolism, and systemic glucose tolerance.

Previously, male mice with global aromatase deficiency exhibited greater body weight than WT controls by 3 months of age [27], although this body weight phenotype was not reproduced in a later study [33]. In the latter study, global aromatase deficiency resulted only in increased gonadal fat mass. In our study, hematopoietic aromatase deficiency did not yield lasting differences in body weight or total adiposity, but a trend toward higher epididymal fat mass was found in WT(ArKO) mice, suggesting that immune cell-specific aromatase deficiency may play a very minor, contributory role in the metabolic phenotype of global knockout mice. Nonetheless, loss of hematopoietic aromatase did not substantially affect adiposity in male mice. Similar body fat mass between WT(WT) and WT(ArKO) mice likely explains the comparable concentrations of circulating adipokines in both groups, in contrast to prior findings of elevated plasma leptin and reduced adiponectin concentrations in male mice with global aromatase deficiency [33]. One possible explanation for these overall null findings is that aromatase expression is high within adipose tissue, with aromatase activity in multiple cell types. Further, estrogens are stored in adipose tissue in the form of fatty acyl esters and estrogen sulfates and can be converted into bioactive estrogens through the activity of lipases and steroid sulfatase, respectively. It is possible, therefore, that estrogen generation through these alternative mechanisms fully compensated for loss of aromatase activity in tissue immune cells. This is particularly plausible as the animals were exposed only to a chow diet, so adipose tissue immune cell infiltration remained low. The metabolic importance of total aromatase activity within adipose tissue is supported by a recent study demonstrating that adipose-specific aromatase overexpression resulted in improved insulin sensitivity [34].

Further, WT(ArKO) mice did not develop time-dependent insulin resistance or hepatic steatosis, as has been observed in male mice with global aromatase deficiency and ascribed specifically to hepatic insulin resistance [27, 33]. Thus, in male mice with global loss of aromatase activity, overt hyperglycemia was seen by 3 months of age in association with increased expression of gluconeogenic genes in liver and, subsequently, increased hepatic triglyceride accumulation [33]. The present study did not reveal any evidence of changes in hepatic energy metabolism or insulin sensitivity, in contrast to these models of global aromatase deficiency in male mice. Therefore, hematopoietic aromatase does not appear to contribute to the hepatic insulin resistance or steatosis evident in global aromatase deficiency. Nonetheless, similar to findings in adipose tissue, we cannot definitively conclude whether the absence of a liver phenotype in the present study is due to a limited physiologic role for aromatase activity in liver immune cells or compensation through estradiol production by alternative pathways.

Plasma concentrations of 17β-estradiol were comparable between WT(WT) and WT(ArKO) mice, indicating that immune cells do not contribute significantly to circulating estrogens, at least via aromatization. Our findings further are consistent with prior work demonstrating similar plasma concentrations of triglycerides and cholesterol in mice with global aromatase deficiency and controls [27], though, in the prior study, global aromatase deficiency led to post-prandial hyperlipidemia after 36 weeks of age.

Our study has several key limitations. Although successful bone marrow engraftment was demonstrated in circulating immune cells, we cannot exclude the possibility that some host tissue immune cell populations persisted or expanded subsequent to transplant. However, unless

the transplanted donor cells are expected to have impaired maturation or function, these donor cells should completely replace host tissue immune cells within 2–4 months after transplant in all metabolic tissues except brain [35]. Future work using labeled bone marrow cells will be important for careful quantitation of transplanted cells in metabolic tissues. Protein analyses also will be important to support the gene expression findings in our study; protein analyses were attempted for this cohort of animals but were not successful due to the methods employed for tissue processing at animal sacrifice. Future work also should include careful assessment of energy expenditure and food intake through calorimetry, particularly as body weight gain in male mice with global aromatase deficiency has been attributed at least in part to reduced volitional activity [9]. As noted previously, since estrogens can be generated within tissues through various pathways, our study cannot comprehensively define the metabolic roles of immune cell-derived estrogens; rather, it only addresses loss of estrogen production specifically through an aromatase-dependent mechanism. Therefore, future studies also should entail measurement of estrogen production through the steroid sulfatase pathway. Our study entailed radiation exposure and bone marrow transplant, which itself confers metabolic effects specifically with regard to reducing body weight gain, inhibiting adipogenesis, and increasing insulin resistance [36, 37]. However, these effects have been demonstrated with high fat feeding to confer nutritional stress, whereas our study employed a regular chow diet [36]. Critically, bone marrow transplant and ionizing radiation do not have a sustained impact on glucose homeostasis in mice exposed to a chow diet [36, 37], suggesting that our primary findings were not confounded by the use of this procedure. Exposure to ionizing radiation also was shown to inhibit proliferation in skeletal muscle satellite cells, although this was not associated with changes in insulin-mediated AKT phosphorylation within skeletal muscle [37]. Nonetheless, the possibility exists that employing bone marrow transplant contributed to our null findings in adipose tissue. Additional work is also needed to determine whether the observed differences in skeletal muscle energy metabolism are due to 17β-estradiol signaling or ligand-independent effects mediated by ERα. Growth factors, cytokines, and other signals have been shown to stimulate unliganded ERα-mediated transcriptional activation and repression [38–40], most commonly through serine phosphorylation [39]. Therefore, changes in ligand-independent ERα function may have contributed to the current findings. Finally, parallel work must be performed in models of high fat feeding and obesity to determine the interaction between metabolic stress and immune cell aromatase activity, as well as in female mice to establish whether sexual dimorphisms exist in the metabolic roles of immune cell-derived estrogens. Importantly, the chow diet employed in our study did not contain any ingredients that would contribute isoflavones or other phytoestrogens, as these have estrogenic properties and can achieve high circulating concentrations when administered through rodent diets [41].

Immune cells have extensive capacity to synthesize and secrete estrogens. These findings demonstrate initial evidence that this facet of immune cell function may play important metabolic roles. Extensive additional studies are needed to delineate the respective intracrine and paracrine effects of immune cell-derived estrogens and to determine how immune cell aromatase activity is regulated within metabolic tissues. Finally, this study contributes to a growing body of evidence demonstrating the importance of estrogens for influencing metabolic health in males.

## Acknowledgments

The authors would like to thank Nilesh Gaikwad and the Gaikwad Steroidomics Lab, LLC, for measurement of tissue estrogen concentrations.

## Author Contributions

**Conceptualization:** Katya B. Rubinow.

**Data curation:** Katya B. Rubinow, Laura J. den Hartigh, Leela Goodspeed, Shari Wang.

**Formal analysis:** Katya B. Rubinow.

**Funding acquisition:** Katya B. Rubinow.

**Investigation:** Orhan K. Oz.

**Methodology:** Orhan K. Oz.

**Resources:** Orhan K. Oz.

**Supervision:** Katya B. Rubinow.

**Writing – original draft:** Katya B. Rubinow.

**Writing – review & editing:** Laura J. den Hartigh, Leela Goodspeed, Shari Wang, Orhan K. Oz.

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
