## [Decision Letter · Decision Letter 0]

29 Oct 2019

PONE-D-19-24315

Aromatase deficiency in hematopoietic cells improves glucose tolerance in male mice through skeletal muscle-specific effects

PLOS ONE

Dear Dr. Rubinow,

Thank you for submitting your manuscript to PLOS ONE. After careful consideration, we feel that it has merit but does not fully meet PLOS ONE’s publication criteria as it currently stands. Therefore, we invite you to submit a revised version of the manuscript that addresses the points raised during the review process. In particular you should supply more data supporting the conclusions which are now only based on statistically insignificant trends in the current results.

We would appreciate receiving your revised manuscript by Dec 13 2019 11:59PM. To enhance the reproducibility of your results, we recommend that if applicable you deposit your laboratory protocols in protocols.io, where a protocol can be assigned its own identifier (DOI) such that it can be cited independently in the future. For instructions see: http://journals.plos.org/plosone/s/submission-guidelines#loc-laboratory-protocols

We look forward to receiving your revised manuscript.

Kind regards,

Michael Bader

Academic Editor

PLOS ONE

Journal Requirements:

3. At this time, we request that you  please report additional details in your Methods section regarding animal care, as per our editorial guidelines: 1) Please provide the number and source of the donor mice 2) Please provide details of animal welfare for both the donor and recipient mice (e.g., shelter, food, water, environmental enrichment) 3) please describe any steps taken to minimize animal suffering and distress, such as by administering analgesics, for the recipient mice 4) please include the method of sacrifice for both the donor and recipient mice and 5) Please describe the post-operative care received by the recipient animals, including the frequency of monitoring and the criteria used to assess animal health and well-being. Thank you for your attention to these requests.

Reviewers' comments:

Reviewer's Responses to Questions

**Comments to the Author**

1. Is the manuscript technically sound, and do the data support the conclusions?

Reviewer #1: No

Reviewer #2: Yes

2. Has the statistical analysis been performed appropriately and rigorously? 

Reviewer #1: Yes

Reviewer #2: Yes

3. Have the authors made all data underlying the findings in their manuscript fully available?

Reviewer #1: Yes

Reviewer #2: Yes

4. Is the manuscript presented in an intelligible fashion and written in standard English?

Reviewer #1: Yes

Reviewer #2: Yes

5. Review Comments to the Author

Reviewer #1: The study by Rubinow et al attempts to investigate the role of estradiol produced by immune cells in regulating metabolism. To achieve this, the authors use a clever approach of performing bone marrow transplantation experiments in mice using WT and aromatase-deficient donors.

The authors clearly demonstrate that cultured immune cells isolated from the peritoneal of mice secrete high levels of 17B-Estradiol and estrone. The phenotype of the WT mice transplanted with bone marrow from ArKO mice however is mild with only statistically significant differences being identified as an increase in fat mass at 8 weeks and a decrease in glucose tolerance at 16 weeks of age.

Unfortunately, the authors have over-stated the data with the first 3 paragraphs of the discussion based on data that is not statistically significant, particularly with regards to the gene expression data in skeletal muscle and the concentrations of 17B-estradiol within skeletal muscle. Throughout the manuscript, the authors use the word “trend” to describe data with P values of >0.05 (eg. P=0.09).

Whilst the premise for this study has the potential to provide important insight into the role of estradiol derived from immune cells in metabolic regulation, the data presented in this paper is premature and requires additional analyses to ensure accurate interpretation of the data.

Other comments:

Why are the data expressed at Mean +/- SD in the text and SEM in the figures? These should be consistent between the text and the figures. SD is appropriate for replicates of cell culture experiments, and SE for experiments consisting of individual animals.

The gene expression data in skeletal muscle could be further supported by protein analyses.

The housekeeping genes Gapdh and Actb have been shown not to be optimal for normalisation of gene expression data as they are not stably expressed. As such, alternative house keeping genes should be used.

Reviewer #2: Estrogen is known to play an important metabolic role in females and have now been acknowledged to have a role in males. Similar to females, loss of estogen results in increased body mass due to increased fat mass, and reduced insulin sensitivity. Thus, having an overall negative effect on the metabolic health of the individual. Additionally, global aromatase deficiency results in a similar metabolic phenotype, but the mechansms for estrogen-based metabolic regulation in males is not completely understood. Therefore, the authors aim to uncover the source of estrogen production and tissue-specific site of action which results in these metabolic defects in males. Thus, the authors aim to examine the metabolic effects of estrogens specfically produced by immune cells. The authors emply bone-marrow transplants from wild-type and aromatase knock-out animals into wild-type animals in order address this question. While the results from this manuscript conflict the results from previous studies the authors do not over interpret their results and are transparent with their study limitations. Minor comments to improve the manuscript are as follows:

1. Please report more specific RT-qPCR methods such as (taqman or syber green): supplementary table with genes analyzed and primer sequences

2. While transcript data is useful, protein data would be more valuable and strengthen the data to support the conclusions that these advantageous metabolic adaptations occurred.

3. The addition of metabolic chamber data would strengthen the manuscript but not required.

6. PLOS authors have the option to publish the peer review history of their article (what does this mean?). If published, this will include your full peer review and any attached files.

Reviewer #1: No

Reviewer #2: No

---

## [Author Response · Author response to Decision Letter 0]

12 Dec 2019

We thank the journal and reviewers for their critique of our paper. Please find below our itemized responses to the journal requirements and reviewers’ comments. 

Journal Requirements:

We have reviewed all style requirements and labeled files as indicated. 

We apologize for use of this phrase. We have removed the phrase in reference to the food intake data (lines 257-258); as the animals were group housed, we only have estimates for food intake data for the individual animals. Although these estimates do not suggest a difference in food intake, we have chosen to omit this altogether in the absence of higher quality data.

This phrase also was used in reference to measurements of adipocyte size. These data are now shown in Figures 8C and 8D.

3. At this time, we request that you please report additional details in your Methods section regarding animal care, as per our editorial guidelines: 1) Please provide the number and source of the donor mice 2) Please provide details of animal welfare for both the donor and recipient mice (e.g., shelter, food, water, environmental enrichment) 3) please describe any steps taken to minimize animal suffering and distress, such as by administering analgesics, for the recipient mice 4) please include the method of sacrifice for both the donor and recipient mice and 5) Please describe the post-operative care received by the recipient animals, including the frequency of monitoring and the criteria used to assess animal health and well-being. Thank you for your attention to these requests.

We apologize for omission of this information and have added the following details to our manuscript:

1) Number and source of donor mice: The breeder mice were from a congenic C57BL6/J strain and provided from an animal colony at the University of Texas-Southwestern maintained by the senior author (OKO); these included 1 male and 2 female mice heterozygous for aromatase deficiency and 1 wild-type male mouse. (lines 113-115)

2) Details of animal welfare: All mice were group housed (4-5 animals per cage, breeding pairs 2 animals per cage) with ad libitum access to food and water. (lines 176-177)

3) Steps to alleviate animal suffering: In order to minimize suffering and risk to animals, fasting times were limited to 4-5 hours prior to metabolic testing, bone marrow transplant and sacrifice were performed under isoflurane anesthesia, and mice were placed on heating pads immediately following fat-water imaging. Adequacy of anesthesia was assessed through monitoring of respiratory rate and toe pinch response. (lines 177-181)

4) Method of sacrifice: 

Donor mice: The morning following irradiation, donor mice were sacrificed by CO2 inhalation and exsanguination, and femurs and tibias were harvested. (lines 143-144)

Recipient mice: Measurement of visceral, subcutaneous, and liver fat volume subsequently was performed using fat-water imaging (study week 22), and animals were sacrificed at study week 24 through cervical dislocation and exsanguination. (lines 172-174)

5) Post-operative care for recipient mice: Recipients of bone marrow transplant were monitored 3-4 times each week for the 4 weeks subsequent to irradiation until marrow engraftment occurred. Mice were monitored for any signs or behaviors indicative of poor health, including hunching, difficulty breathing, or listlessness, and any concerning signs or behaviors were reported to veterinary medicine. (lines 130-134)

Reviewers' comments:

Reviewer's Responses to Questions

Comments to the Author

Reviewer #1: The study by Rubinow et al attempts to investigate the role of estradiol produced by immune cells in regulating metabolism. To achieve this, the authors use a clever approach of performing bone marrow transplantation experiments in mice using WT and aromatase-deficient donors.

The authors clearly demonstrate that cultured immune cells isolated from the peritoneal of mice secrete high levels of 17B-Estradiol and estrone. The phenotype of the WT mice transplanted with bone marrow from ArKO mice however is mild with only statistically significant differences being identified as an increase in fat mass at 8 weeks and a decrease in glucose tolerance at 16 weeks of age.

Unfortunately, the authors have over-stated the data with the first 3 paragraphs of the discussion based on data that is not statistically significant, particularly with regards to the gene expression data in skeletal muscle and the concentrations of 17B-estradiol within skeletal muscle. Throughout the manuscript, the authors use the word “trend” to describe data with P values of >0.05 (eg. P=0.09).

We appreciate the reviewer’s approval of our approach. Although there was disagreement between reviewers regarding the stringency of our interpretation of the data, we appreciate these comments and did not intend to overstate the findings. We agree the phenotype is mild but nonetheless think the results are strengthened by the consistency between the enhancement of systemic glucose tolerance and gene expression data in skeletal muscle in the WT(ArKO) group. Additionally, we believe the biological significance of our findings is supported by the fact that changes were observed in skeletal muscle but not liver or adipose. To better avoid overstating our findings, we have made the following revisions: 

We have changed the text in our Results section to state that epididymal adipose tissue mass did not differ significantly between groups (lines 272-275), particularly as this finding was found only on post-mortem tissue weight analysis and not corroborated by our fat-water imaging data. 

We also have modified our reporting of gene expression data to present p-values and focus solely on the genes that differed significantly between groups with the exception of Esr1, which we think may be important for interpreting the results collectively. These changes have been made to Abstract (lines 31-32) and the Results section, where mention of the genes Irs1, Tfam, and Cptb1b have been removed from the text (lines 332-342). 

With regard to the skeletal muscle concentrations of 17β-estradiol, we now have presented the figures as scatterplots, in order to better depict that much of the variation in tissue concentrations in the WT(ArKO) group was driven by a single outlier. Although some choose to exclude outliers >2 SD from the mean, we opted instead to show all data but add a sensitivity analysis demonstrating a significant between-group difference with exclusion of this single outlier (lines 369-373). Nonetheless, as this outlier did introduce substantial variation into the data, we explicitly state that the between-group difference did not achieve statistical significance (line 368), and we reiterate in the discussion that the between-group difference in skeletal muscle estradiol concentrations did not achieve significance when all animals were included in the analysis (lines 439-441):

Notably, although tissue estradiol concentrations differed by roughly 1/3, this finding did not quite achieve statistical significance and requires verification in a future study powered for this endpoint.

Whilst the premise for this study has the potential to provide important insight into the role of estradiol derived from immune cells in metabolic regulation, the data presented in this paper is premature and requires additional analyses to ensure accurate interpretation of the data.

We think the data are not premature as related to sample size and the publication philosophy of the journal. Rather, the data reflect the magnitude of biological effects in our experimental design. We have rephrased our interpretation toward a much more conservative interpretation. Further, we think our studies were carried out in a technically sound way, a requirement of the journal.

Other comments:

Why are the data expressed at Mean +/- SD in the text and SEM in the figures? These should be consistent between the text and the figures. SD is appropriate for replicates of cell culture experiments, and SE for experiments consisting of individual animals.

We appreciate this suggestion. Cell culture data are now shown as mean ± SD in the figure as well as in the text, and all other data are now shown as mean ± SEM. The statistical analysis section of the Methods has been updated accordingly (lines 238-240).

The gene expression data in skeletal muscle could be further supported by protein analyses.

We agree that protein analyses are important for supporting our gene expression data. Consequently, we extracted protein from skeletal muscle and performed Western blotting for ERα, aromatase, and GLUT4, as well as β-tubulin and β-actin for normalization. Unfortunately, we only had formalin-fixed tissue remaining, and despite the use of 2 different protein extraction methods, we failed to get a signal for any of the probed proteins, including β-tubulin and β-actin. We very much regret that we are unable to include these additional protein analyses in our paper. 

The housekeeping genes Gapdh and Actb have been shown not to be optimal for normalisation of gene expression data as they are not stably expressed. As such, alternative house keeping genes should be used.

Thank you for this suggestion. Gene expression in epididymal and inguinal adipose tissue has been re-analyzed using the geometric mean of Nono and β-2-microglobulin (B2m) rather than Gapdh for normalization. For the skeletal muscle gene expression data, additional genes for normalization were analyzed, including B2m and 18s. However, when all of the normalization genes were compared, β-actin demonstrated the most stable expression. Therefore, our further work to identify an optimal gene for normalization supported continued use of β-actin as our housekeeping gene for this tissue.

Reviewer #2: Estrogen is known to play an important metabolic role in females and have now been acknowledged to have a role in males. Similar to females, loss of estrogen results in increased body mass due to increased fat mass, and reduced insulin sensitivity. Thus, having an overall negative effect on the metabolic health of the individual. Additionally, global aromatase deficiency results in a similar metabolic phenotype, but the mechanisms for estrogen-based metabolic regulation in males is not completely understood. Therefore, the authors aim to uncover the source of estrogen production and tissue-specific site of action which results in these metabolic defects in males. Thus, the authors aim to examine the metabolic effects of estrogens specifically produced by immune cells. The authors employ bone-marrow transplants from wild-type and aromatase knock-out animals into wild-type animals in order address this question. While the results from this manuscript conflict the results from previous studies the authors do not over interpret their results and are transparent with their study limitations. Minor comments to improve the manuscript are as follows:

1. Please report more specific RT-qPCR methods such as (taqman or syber green): supplementary table with genes analyzed and primer sequences.

We appreciate this suggestion. Greater detail regarding the PCR methods employed has been added to the Methods section (lines 206-214), and Table 1 has been inserted (line 215) and lists the accession numbers for all primer sets used in the analyses. 

2. While transcript data is useful, protein data would be more valuable and strengthen the data to support the conclusions that these advantageous metabolic adaptations occurred.

We agree with this reviewer and have performed additional protein analyses as described above in response to Reviewer #1. Despite multiple attempts to extract protein from our formalin-fixed tissues, we are unable to provide protein data for the manuscript.

3. The addition of metabolic chamber data would strengthen the manuscript but not required.

We also agree with this suggestion. Due to limited funding, we were unable to include metabolic chamber analyses in this study. Although we did not observe a body weight or body composition phenotype in our WT(ArKO) animals, we agree that metabolic chamber data will be necessary in a future study to comprehensively define the metabolic phenotype of mice with hematopoietic aromatase deficiency.

---

## [Decision Letter · Decision Letter 1]

19 Dec 2019

PONE-D-19-24315R1

Aromatase deficiency in hematopoietic cells improves glucose tolerance in male mice through skeletal muscle-specific effects

PLOS ONE

Dear Dr. Rubinow,

Thank you for submitting your manuscript to PLOS ONE. After careful consideration, we feel that it has merit but does not fully meet PLOS ONE’s publication criteria as it currently stands. Therefore, we invite you to submit a revised version of the manuscript that addresses the points still raised by reviewer 1.

We would appreciate receiving your revised manuscript by Feb 02 2020 11:59PM. To enhance the reproducibility of your results, we recommend that if applicable you deposit your laboratory protocols in protocols.io, where a protocol can be assigned its own identifier (DOI) such that it can be cited independently in the future. For instructions see: http://journals.plos.org/plosone/s/submission-guidelines#loc-laboratory-protocols

We look forward to receiving your revised manuscript.

Kind regards,

Michael Bader

Academic Editor

PLOS ONE

Reviewers' comments:

Reviewer's Responses to Questions

**Comments to the Author**

1. If the authors have adequately addressed your comments raised in a previous round of review and you feel that this manuscript is now acceptable for publication, you may indicate that here to bypass the “Comments to the Author” section, enter your conflict of interest statement in the “Confidential to Editor” section, and submit your "Accept" recommendation.

Reviewer #1: (No Response)

2. Is the manuscript technically sound, and do the data support the conclusions?

Reviewer #1: Yes

3. Has the statistical analysis been performed appropriately and rigorously? 

Reviewer #1: Yes

4. Have the authors made all data underlying the findings in their manuscript fully available?

Reviewer #1: Yes

5. Is the manuscript presented in an intelligible fashion and written in standard English?

Reviewer #1: Yes

6. Review Comments to the Author

Reviewer #1: The authors have attempted to address each of the editor’s and reviewer’s concerns and is greatly improved as a result, particularly with regards to the over-interpretation of non-significant data in the first version of the manuscript.

Minor comments;

It is worth noting in the text, that the skeletal muscle gene expression data was expressed relative to beta-actin as it was demonstrated to be the most stably expressed housekeeping gene of those analysed.

It is also worth acknowledging in the text that the gene expression data in skeletal muscle could be further supported by protein analyses, however, this was not possible using the current cohort of samples due to the method in which the tissues were processed.

Line 342 – The use of the word “trend” should be avoided to describe data with a P value of P=0.09. The description of these data could be re-worded to state that the mean value of ERalpha mRNA was higher in the WT(ArKO) mice than the WT mice however this did not reach statistical significance (P=0.09).

Lines 367-368 and Lines 439-441 – Similarly, this sentence should be reworded to state that the mean 17B estradiol levels were lower in WT(ArKO) mice, but this did not reach statistical significance P=0.08. The word “quite” in this sentence implies a P value close to 0.05 (ie. such as 0.06), so should be removed.

7. PLOS authors have the option to publish the peer review history of their article (what does this mean?). If published, this will include your full peer review and any attached files.

Reviewer #1: No

---

## [Author Response · Author response to Decision Letter 1]

27 Dec 2019

We thank the reviewer for additional suggestions for our paper. Please find below our itemized responses to the reviewer’s comments. 

Comments to the Author

Reviewer #1: The authors have attempted to address each of the editor’s and reviewer’s concerns and is greatly improved as a result, particularly with regards to the over-interpretation of non-significant data in the first version of the manuscript.

Minor comments;

It is worth noting in the text, that the skeletal muscle gene expression data was expressed relative to beta-actin as it was demonstrated to be the most stably expressed housekeeping gene of those analysed.

We agree that this additional information is important to include and have added the following sentence to our Methods section (lines 209-211):

Expression of 3 candidate housekeeping genes was measured for each tissue, and the genes exhibiting the most stable tissue-specific expression were utilized for data normalization.

It is also worth acknowledging in the text that the gene expression data in skeletal muscle could be further supported by protein analyses, however, this was not possible using the current cohort of samples due to the method in which the tissues were processed.

We now include this limitation of our study in our discussion, as follows (lines 480-482):

Protein analyses also will be important to support the gene expression findings in our study; protein analyses were attempted for this cohort of animals but were not successful due to the methods employed for tissue processing at animal sacrifice. 

Line 342 – The use of the word “trend” should be avoided to describe data with a P value of P=0.09. The description of these data could be re-worded to state that the mean value of ERalpha mRNA was higher in the WT(ArKO) mice than the WT mice however this did not reach statistical significance (P=0.09).

We appreciate this comment, and the wording of this sentence has been amended as suggested (lines 336-338):

In WT(ArKO) mice, mean expression of ERα (Esr1) was higher relative to WT(WT) mice, although this difference did not achieve statistical significance (p=0.09).

Lines 367-368 and Lines 439-441 – Similarly, this sentence should be reworded to state that the mean 17B estradiol levels were lower in WT(ArKO) mice, but this did not reach statistical significance P=0.08. The word “quite” in this sentence implies a P value close to 0.05 (ie. such as 0.06), so should be removed.

The word ‘quite’ has been removed in both of these places, as advised (lines 362 and 434).

---

## [Editor Report · Decision Letter 2]

31 Dec 2019

Aromatase deficiency in hematopoietic cells improves glucose tolerance in male mice through skeletal muscle-specific effects

PONE-D-19-24315R2

Dear Dr. Rubinow,

We are pleased to inform you that your manuscript has been judged scientifically suitable for publication and will be formally accepted for publication once it complies with all outstanding technical requirements.

With kind regards,

Michael Bader

Academic Editor

PLOS ONE
---

## [Editor Report · Acceptance letter]

6 Jan 2020

PONE-D-19-24315R2 

Aromatase deficiency in hematopoietic cells improves glucose tolerance in male mice through skeletal muscle-specific effects 

Dear Dr. Rubinow:

I am pleased to inform you that your manuscript has been deemed suitable for publication in PLOS ONE. Congratulations! Your manuscript is now with our production department. 

With kind regards,

on behalf of

Prof. Michael Bader 

Academic Editor

PLOS ONE